# Six-Month Follow-Up after Vaccination with BNT162b2: SARS-CoV-2 Antigen-Specific Cellular and Humoral Immune Responses in Hemodialysis Patients and Kidney Transplant Recipients

**DOI:** 10.3390/pathogens11010067

**Published:** 2022-01-05

**Authors:** Simone Cosima Boedecker-Lips, Anja Lautem, Stefan Runkel, Pascal Klimpke, Daniel Kraus, Philipp Keil, Stefan Holtz, Vanessa Tomalla, Paul Marczynski, Christian Benedikt Boedecker, Peter Robert Galle, Martina Koch, Julia Weinmann-Menke

**Affiliations:** 1Department of Nephrology, I. Department of Medicine, University Medical Center Mainz, Johannes Gutenberg University, D 55131 Mainz, Germany; SimoneCosima.Boedecker-Lips@unimedizin-mainz.de (S.C.B.-L.); Pascal.Klimpke@unimedizin-mainz.de (P.K.); Daniel.Kraus@unimedizin-mainz.de (D.K.); Philip.Keil@unimedizin-mainz.de (P.K.); Stefan.Holtz@unimedizin-mainz.de (S.H.); Vanessa.Tomalla@unimedizin-mainz.de (V.T.); Paul.Marczynski@unimedizin-mainz.de (P.M.); 2Department of General, Visceral and Transplantation Surgery, University Medical Center Mainz, Johannes Gutenberg University, D 55131 Mainz, Germany; anja.lautem@unimedizin-mainz.de (A.L.); Christian.Boedecker@unimedizin-mainz.de (C.B.B.); martina.Koch@unimedizin-mainz.de (M.K.); 3Blood Transfusion Center, University Medical Center Mainz, Johannes-Gutenberg University, D 55131 Mainz, Germany; Stefan.Runkel@unimedizin-mainz.de; 4Department of Internal Medicine, University Medical Center Mainz, Johannes Gutenberg University, D 55131 Mainz, Germany; peter.galle@unimedizin-mainz.de; 5Research Center of Immunotherapy (FZI), University Medical Center Mainz, Johannes Gutenberg University, D 55131 Mainz, Germany

**Keywords:** COVID-19, transplantation, immunosuppression, end-stage kidney disease

## Abstract

Hemodialysis patients (HDP) and kidney transplant recipients (KTR) have a high risk of infection with SARS-CoV-2 with poor clinical outcomes. Because of this, vaccination of these groups of patients against SARS-CoV-2 is particularly important. However, immune responses may be impaired in immunosuppressed and chronically ill patients. Here, our aim was to compare the efficacy of an mRNA-based vaccine in HDP, KTR, and healthy subjects. Design: In this prospective observational cohort study, the humoral and cellular response of prevalent 192 HDP, 50 KTR, and 28 healthy controls (HC) was assessed 1, 2, and 6 months after the first immunization with the BNT162b2 mRNA vaccine. Results: After 6 months, 97.5% of HDP, 37.9% of KTR, and 100% of HC had an antibody response. Median antibody levels were 1539.7 (±3355.8), 178.5 (±369.5), and 2657.8 (±2965.8) AU/mL in HDP, KTR, and HC, respectively (*p* ≤ 0.05). A SARS-CoV-2 antigen-specific cell response to vaccination was found in 68.8% of HDP, 64.5% of KTR, and 90% of HC. Conclusion: The humoral response rates to mRNA-based vaccination of HDPs are comparable to HCs, but antibody titers are lower. Furthermore, HDPs have weaker T-cell response to vaccination than HCs. KTRs have very low humoral and antigen-specific cellular response rates and antibody titers, which requires other vaccination strategies in addition to booster vaccination.

## 1. Introduction

Patients with end-stage kidney disease (ESKD) and kidney transplant recipients (KTR) have a high risk of infection due to their frequent visits to a clinic accompanied by a high risk of mortality after infection with SARS-CoV-2 [1]. Therefore, these patients were prioritized for SARS-CoV-2 vaccination in Spring 2021 in Germany and other countries. Antibody data suggest that the immune response to SARS-CoV-2 vaccination under immunosuppression is low [2,3,4,5]. In addition, it is known that patients with ESKD or after solid-organ transplantation have a significantly poorer response to previous vaccines, such as influenza or hepatitis B [6,7,8]. However, published data have shown an unexpectedly strong humoral response, and it has also shown a less studied cellular vaccination response after mRNA-based vaccination in patients with dialysis treatment. So far, humoral vaccination response after the second vaccination with BNT162b2 could be detected in up to 70–97.5% of dialysis patients [9,10,11,12,13]. In contrast, patients after kidney transplantation unfortunately displayed a significant weaker vaccination response. Humoral response against SARS-CoV-2 has been observed in 22–48% of kidney transplant recipients (KTR) after the second vaccination [2,3,4,5,10,14]. Due to the poor vaccination response in all solid-organ transplant recipients, a third vaccination dose was recommended very early by the French National Authority for Health in France. In a cohort of 101 organ transplant patients, antibodies were detected in 40% after two doses of the mRNA vaccine and in 68% after a third dose [3]. Likewise, in KTR, two doses elicited an antibody response in only 32% of patients, whereas 55% had an antibody response after the third dose [15]. Hall et al. were also able to show the positive effect of a third dose of an mRNA vaccine in transplanted patients in a randomized trial [16].

In addition to antibody response, T-cell responses are also likely to play a key role. T-cells, with their antiviral signatures, have the ability to control viral replication and host disease, and play an important role in SARS-CoV-2 infection [17]. SARS-CoV-2 antigen-specific T effector cell response in healthy individuals is also well documented after vaccination [18]. Further evidence of the cellular vaccination response was described by Geers et al. whose work demonstrated that SARS-CoV-2 variants of concern can escape the humoral, but not the cellular response to vaccination or following COVID-19 convalescence [19]. Previously published work by Sattler et al. demonstrated that 8 days after the first vaccination, spike-specific T helper cells are detected in most transplanted patients (flow cytometry). In comparison to HDP and a healthy control, these were significantly reduced, so that subsequently a clear impairment of effector cytokine production could also be demonstrated [20]. Hugo et al. also examined, besides the humoral vaccination response, the cellular response after vaccination with mRNA vaccines. In the study SARS-CoV-2 antigen-specific T effector cells were examined by an IFN-γ release assay and by using flow cytometry. Thereby, a cellular immune response by vaccination could be detected in 78% in HDP and by 30% in KTR [13]. Similar data were published by Bertrand et al. and Cucchiari et al., who were able to detect in comparison a better cellular response in their patient groups (100% cellular vaccination response in HDP and 35–51% in KTR) [21,22]. These results seem disillusioning for patients after kidney transplantation; however, so far, not many data are available that focus on cellular vaccination response in this high-risk population. 

In addition to the focused research on the development of a vaccine and the investigation of the vaccination response of different vulnerable patient groups in the last year, a large research focus has been placed on genetics, the possible influence on the infection rate in different countries, and the different disease courses of COVID-19. A special focus was placed on the connection between the HLA genotype and the severity of COVID-19 disease. In several large studies, it was possible to identify HLA alleles that could be associated with a severe to lethal course of COVID-19. In particular, A*11, B44, C*1, DRB1*8 and DQB1*4 should be mentioned here [23,24,25,26]. 

All together, it remains unclear how long vaccine protection lasts at both the humoral and cellular level in both kidney transplant and dialysis patients [5]. The collective of dialysis patients concerning the few data published to date indicates rapidly declining antibody vaccination response [27,28,29]. In addition, a decrease in seroconversion was also demonstrated by Davidovic [28]. All the more important is the question, to what extent does a cellular vaccination response persist? In this prospective study, we present the results of the humoral and cellular immune response measured by the SARS-CoV-2-specific antibody assay and the SARS-CoV-2 antigen-specific cellular response by IFN-γ/IL-2 ELISpot assay in dialysis and kidney transplanted patients after vaccination with BNT162b2. In addition, we examined whether the HLA genotype can be associated with the vaccination response.

## 2. Results

### 2.1. Patient Characteristics

In total, 270 patients were included in our study. Of these, 192 patients required dialysis and 50 patients had a previous kidney transplantation. In addition, a healthy control group was included (*n* = 28). All subjects received a SARS-CoV-2 vaccination with the BNT162b2 vaccine with an interval of 4 weeks between the first and second dose. The vaccination was well tolerated by all study participants and no serious side effects were documented. SARS-CoV-2-specific antibodies were determined in all study participants 1 month after the first (V1), 1 month after the second vaccination (V2), and 6 months after first vaccination (V3). Additionally, in one subgroup, SARS-CoV-2 antigen-specific cellular response was measured by an Interferon-γ (IFN-γ) and Interleukin2 (IL-2) ELISpot assay. The study cohort characteristics are summarized in Table 1.

As expected, the group of HDP has a higher age (66.9 years) than the KTR (51.2 years); the age of the control group is comparable to that of the transplant patients (54.1 years). In terms of gender distribution, the group of HDP is comparable to the group of KTR. With regard to the comorbidities, there is a 30% presence of overweight (BMI: 26–31) in all study groups; in addition, 21.4% of the HDP group have severe obesity. Other comorbidities (diabetes > hypertension > heart disease > chronic lung disease > gastrointestinal/liver disease > tumor disease) are comparably high in the collectives of HDP and KTR. In the control group, there are almost no pre-existing conditions (see Table 1).

On average, KTR had been transplanted 82 (±74.3) months ago. Previously, these patients were on dialysis for an average of 61.8 (±44.7) months. In 45 of these patients a triple immunosuppression was used and only five received a dual immunosuppression (see Table 2). Moreover, the HDP included in the study have been on dialysis for 54 (±42.8) months. Of the 192 patients, 17 patients (8.8%) were immunosuppressed during the study period by immunosuppressive therapy for an autoimmune disease (*n* = 3, all 3 patients received Rituximab) or due to an active tumor disease (*n* = 14).

### 2.2. Humoral Immune Response by Measurement of SARS-CoV-2 Antibodies after Vaccination with BNT162b2

The humoral response to vaccination was measured by anti-spike IgG (against the receptor-binding domain (RBD) of the spike protein) antibodies. In patients after kidney transplantation, SARS-CoV-2-specific anti-spike antibodies could be detected in 12.5% in V1 and in 39.5% in V2 of the patients using the defined cutoff >50 AU/mL, as provided by the manufacturer. In comparison, HDP showed a vaccination response of 83.0% after V1 and 98.0% in V2 (Figure 1a). However, compared to the healthy subjects, there was a clear difference in the measurements of V1 in HDP, with a vaccination response of only 83.0% compared to a vaccination response in V1 of 100% in healthy subjects. This suggests a reduced or perhaps slower vaccination response, as the measured anti-spike antibody titers also differ significantly in their level. HDP have titers of anti-spike IgG with a median of 615.2 (±3089.9) AU/mL in V1 compared to 3157.5 (±4806.4) AU/mL in V2. HDP demonstrates a significantly reduced anti-spike antibody level 3157.5 (±4806.4) AU/mL in comparison to the healthy controls 11929.5 (±8574.1) at V2 (*p* < 0.05). KTR demonstrated an even worse vaccination response with significantly reduced anti-spike IgG (V2: 294.0 ± 667.0) compared with either HDP or the healthy control group (*p* < 0.05). Furthermore, our data confirm the significantly weaker response to vaccination after kidney transplantation; however, in our cohort, despite the use of triple immunosuppression with an antimetabolite/mTor inhibitor in the majority of patients, seroconversion was 39.5% at V2. Remarkable in all study groups was the large standard deviation, which represents large ranges in individual titers.

Six months after the first vaccination, HDP demonstrated a significant decline in anti-spike antibody titers compared to 1 month after the second vaccination (3157.5 vs. 1539.7 AU/mL; see Figure 1b), but with a continued stable seroconversion rate of 97.5%. Interestingly, a significant decrease in anti-spike antibody titers was also observed in the healthy control group, which was much more pronounced compared to HDP. Here, an anti-spike antibody titer of 2657.8 AU/mL could be measured on average after 6 months, while the titer in V2 was 11,929.5 AU/mL. In addition, a decrease of the titer could also be measured in the transplanted patients, which did not reach significance (*p* = 0.57; 299.9 AU/mL in V2 vs. 178.5 AU/mL in V3). However, the seroconversion (anti-spike antibody titer above 50 AU/mL) at time-point V3 was still present in 37.9% of KTR patients in comparison to 39.5 after V2.

### 2.3. SARS-Cov-2 Antigen-Specific Cellular Response Following Vaccination

Because of a quite relevant number of patients who did not develop a humoral immune response to SARS-CoV-2 vaccination, we simultaneously investigated the presence of a cellular immune response to SARS-CoV-2 vaccination (see Table 3). Using the IFN-γ /IL-2 ELISpot assay, the detection of SARS-CoV-2 antigen-specific cellular response was already possible at time-point V1 in the healthy control group in five out of the five patients studied. In comparison, SARS-CoV-2 antigen-specific cellular response was detectable in only 29% of the HDP at V1. However, at time-point V2, a significant increase in the SARS-CoV-2 antigen-specific cellular response of 87% (reactivity only for IL-2: 25%, and reactivity for IFN-γ and IL-2: 62%) was found in this patient population. Six months after the first vaccination, the SARS-CoV-2 antigen-specific cellular response can still be detected in 68.75% (only IFN-γ: 6.25%, only IL-2: 18.75%, and reactivity for IFN-γ and IL-2: 43.74%) of patients in the cohort of HDP. In summary, a markedly reduced SARS-CoV-2 antigen-specific cellular immune response is evident, especially for the IFN-γ positive SARS-CoV-2 antigen-specific cellular immune response. 

In contrast, a SARS-CoV-2 antigen-specific cellular response was detected in 33.3% (only IFN-γ: 11% and only IL-2: 22%) of patients at V1 and 60% (only IFN-γ: 0%, only IL-2: 12%, and reactivity for IFN-γ and IL-2: 48%) of patients at V2 after kidney transplantation (see Table 3). Interestingly, no double-positive SARS-CoV-2 antigen-specific cells were detectable in the KTR group after the first vaccination, but after the second vaccination they were detectable in 48% of KTR patients (see Table 3). In V3, 64.5% (only IFN-γ: 0%, only IL-2: 41%, and reactivity for IFN-γ and IL-2: 23%) of the examined patients demonstrated a SARS-CoV-2 antigen-specific cellular response. It should be highlighted in the individual comparison that in 12 patients who showed an antigen-specific cellular vaccination response in V2, 11 of 12 continued to have a detectable response in V3 (only IL-2: *n* = 7, only IFN-γ: *n* = 0, and IL-2 and IFN-γ: *n* = 4). Looking at the overall vaccination response of KTR at V2, 39.5% of the patients presented with positive anti-spike antibodies but 60% presented with a cellular vaccination response. It is interesting to note here that 32% of these patients (20% cellular only and 12% humoral only) presented either a humoral or antigen-specific cellular immune response to vaccination, so that overall, 72% of the KTR studied had either a humoral and/or cellular vaccine response detectable at time-point V2. This overall vaccination response also remained at 70.5% in V3, with, however, falling antibody titers and, in particular, a clearly visible reduction in the IFN-γ positive antigen-specific cellular immune response.

Compared to the healthy control, which already presented a 100% cellular vaccination response at V1 in all investigated participants, both HDP and KTR demonstrated a significantly delayed and weaker cellular vaccination response. However, the control group also demonstrated a decrease in antigen-specific cellular immunity (only 9 out of 10 patients continued to show a positive detection), and, in particular, the control group also demonstrated a reduction in the IFN-γ positive antigen-specific cellular immune response.

### 2.4. Impact of Cofactor on Vaccination Response

After evaluating the humoral vaccination response in relation to age in all three groups (HDP, KTR, and HC), no significant difference could be found, which is shown in Figure 2a–c. However, a trend for poor humoral vaccination response is found in >60-year-old KTR, which cannot be replicated in the control group. The collective of HDP could be divided into four age groups because of the higher age of the total cohort (Figure 2b). HDP at the age between 60–80 years had a comparable anti-spike antibody titer to patients <60 years. However, in HDP patients >80 years, there is a trend towards lower anti-spike antibody titers at V2 and V3. In addition, no significant difference in humoral vaccination response related to gender could be demonstrated (Figure 2d–f).

Investigating the different immunosuppressive regimens, we can demonstrate that patients receiving dual immunosuppression after kidney transplantation have a better humoral vaccination response compared to patients under triple immunosuppression but with a very small number of patients treated with dual immunosuppression (*n* = 5; Appendix A). Furthermore, no relevant difference in vaccination response depending on the time of transplantation could be demonstrated in our patient collective (see Appendix A). 

With regard to the HLA type, we also investigated whether the humoral vaccination response is related to the HLA genotype of the patients. After differentiating the HLA types, no correlation between an HLA allele and a reduced or improved vaccination response could be observed. Furthermore, no correlation for a reduced vaccination response could be observed between the HLA alleles known from the literature (A*11, B44, C*1 and DRB1*8), which could be associated with a very severe or lethal COVID-19 disease (see Appendix A).

During the study period, in one HDP and one patient of the control group, a new onset of SARS-CoV-2 infection was detected. In the dialysis patient, SARS-CoV2 infection had been detected in the period between the first and second vaccination, but without severe COVID-19 symptoms. The patient in the control group had an incidental finding, so that an asymptomatic infection between the two vaccinations can be assumed. Between V2–V3, SARS-CoV-2 infections were detected in four patients in the HDP collective. Three of the patients could be treated at home with mild symptoms, one patient, 82 years of age, had to be treated in hospital and was discharged home after 3 weeks. These patients were excluded from further study. 

In summary, HDP patients demonstrate very good humoral and antigen-specific cellular vaccination responses, whereas we demonstrate that this is less effective in KTR even when taking in account the humoral or antigen-specific cellular responses. Of particular relevance, however, is the fact that in both HDP and KTR, comparable to healthy controls, the anti-spike antibody titers drop only 6 months after the first vaccination; however, these patients have a significantly lower initial level of anti-spike antibodies from the beginning.

## 3. Discussion

In summary, vaccination with BNT162b2 conveyed an excellent humoral and antigen-specific cellular immune response in HDP, comparable to healthy persons. These positive results are of great importance for the treatment of HDP in dialysis centers. On the other hand, our study also confirms the significantly weaker response to vaccination of immunosuppressed patients after kidney transplantation with significantly reduced vaccination anti-spike antibody titers. However, we did detect antigen-specific cellular responses in a considerable number of KTR. Looking at the six-month follow-up of the humoral and antigen-specific cellular vaccination response, there is a clear decrease in the humoral vaccination response measured by anti-spike antibodies, as well as in the antigen-specific cellular vaccination response, especially through a decrease in the detection of the IFN-γ-specific cellular immune response in all study collectives. However, it is currently unclear what level of circulating anti-spike antibodies and what number of SARS-CoV-2 antigen-specific cells are required to protect from infection with SARS-CoV-2 (or from severe disease following an infection). Of importance, though, increasing evidence is demonstrating that neutralizing capacity is a reliable correlation of protection against SARS-CoV-2 infection [30,31,32]. In addition, there is increasing evidence that anti-spike antibody titers correlate with neutralizing capacity [33].

Compared to previously published data, we detected not only SARS-CoV-2 antigen-specific INF-γ-producing cells, but also IL-2-producing cells in our study, which could be responsible for the significantly better cellular SARS-CoV-2 antigen-specific vaccine response of our cohort [12,13,20,21,34]. This is true especially in KTR, due to the fact that these patient cohorts seem to have a better vaccine response in regard to IL-2-producing cells. Dolff et al., to our knowledge, are the only ones to date that also used IL-2-producing T-cells to demonstrate this better response of IL-2-producing T-cells compared to IFN-γin KTR in their work. It should be highlighted here that Dolff et al. demonstrated a cellular vaccine response in seven out of seven patients after the second vaccination, although the timing of the measurement is not clear [35]. 

Looking at the humoral vaccination response in relation to immunosuppression, there is a significantly higher response under a dual immunosuppression regimen than under a triple drug immunosuppression. These results were already demonstrated in the work of Stumpf et al. [13]. It is doubtful that this will have an effect on the long-term therapy of transplanted patients, as the trade-off between infectious diseases on the one hand and a transplant rejection on the other hand often does not have options in reducing immunosuppression. For this reason, it should be considered whether there are other ways to improve vaccine response in this vulnerable patient group. 

Besides immunosuppression, risk factors for weaker vaccine responses have been repeatedly discussed in the past. In this context, the large standard deviation in the humoral vaccination response, which represents large ranges of individual anti-spike antibody titers should also be mentioned, which also suggests a different individual vaccination response. If we look at the significantly weaker vaccination response of the dialysis patients compared to the healthy controls in our study, this raises the question of what the decisive factor is. On the one hand, dialysis patients are known to have an impaired immune system due to alterations in innate and adaptive immunity. On the other hand, our study population of dialysis patients was significantly more likely to suffer from comorbidities, such as cardiovascular disease, hypertension, diabetes mellitus, or obesity. These comorbidities were also associated with increased mortality in COVID-19 disease [1,36]. Interestingly, our data demonstrated no significant correlation between advanced age and reduced humoral vaccine response. Only KTR over 60 years and dialysis patients older than 80 years demonstrated a tendency for a poorer vaccination response. At the same time, it has been shown in other research groups that advanced age was associated with a reduced humoral vaccination response even in hemodialysis patients and healthy ones [8,9]. Likewise, we demonstrated that gender does not affect vaccination response, although some publications reported more severe disease courses in men compared to women [37].

Due to the significantly reduced vaccination response in KTR, it has already been demonstrated several times that a third mRNA vaccination against SARS-CoV-2 leads to a better vaccination response [3,15,16]. This was demonstrated for the humoral vaccination response, as well as the stronger antigen-specific cellular vaccination response published by Massa et al. [38]. Due to this, a booster vaccination has already been firmly implemented for transplanted patients. In review of the available studies, in Germany the STIKO (Standing Committee on Vaccination) officially recommended booster vaccination for patients with immunodeficiencies on September 30, including patients after transplantation and patients requiring dialysis [39]. In severely immunosuppressed patients such as KTR, booster vaccination is now recommended as early as 4 weeks after the second vaccination, whereas dialysis patients should receive their booster vaccination after 6 months. This recommendation is supported by our data, as we see a significantly lower vaccination response in KTR with especially significant lower anti-spike antibodies, so that in a high number of patients a sufficient protection against a severe SARS-CoV-2 course is unclear and not expected. Because of this, booster vaccination is already useful in the short-term interval and should be carried out as recommended. In addition, it is now necessary to consider what the further vaccination strategies are for KTR who do not achieve seroconversion even after the third vaccination or who have only low anti-spike antibodies after a third vaccination. However, initial data from case series seem to indicate that there is no better or only a slightly better seroconversion after a fourth compared to a third vaccination with, however, an increase in anti-spike antibody titers in patients who showed weak serconversion [40,41]. Fortunately, it is comforting to know that no effects of vaccination on antibody-mediated rejection (including no increased incidence of denovo donor-specific antibody (DSA)) or graft function have been observed to date [42]. Another interesting aspect in this context could be that a significantly higher humoral immunogenicity of the SARS-CoV-2 vaccine mRNA-1273 (Moderna) could be demonstrated compared to the BNT162b2 vaccine (Pfizer-BioNTech) [43]. In view of this, consideration should be given to whether booster vaccinations should be exclusively administered with mRNA-1273 to immunosuppressed patients in the future. Furthermore, it should be ensured that all patients on the waiting list for kidney transplantation have sufficient vaccination protection so that these patients are immunized if an organ is available. 

With regard to HDP, our data show a significant drop in the vaccination titers after 6 months, which is confirmed by the few data published so far [21,22]; this means that at this time a booster vaccination also appears indispensable and has been carried out throughout Germany. However, it is very reassuring that after 6 months there is no decrease in seroconversion after vaccination and also that the antigen-specific cellular vaccination response in this collective has only slightly decreased. This predicts that booster vaccinations will respond very well in HDP. In addition, the very good response of HDP to the SARS-CoV-2 vaccination can be considered as very promising when compared to the previous significantly reduced immune responses to hepatitis B or influenza vaccinations. This was repeatedly attributed to uremia, low hemoglobin levels, or the age of the patients [3,4]. In contrast, the nucleoside-modified RNA (modRNA) vaccine, which encodes the full-length SARS-CoV-2 spike protein, does not appear to be affected by the previously mentioned risk factors of conventional inactivated vaccines. Accordingly, the further development of mRNA vaccines for HDP might also be a promising option with regard to other infectious diseases.

To the best of our knowledge, however, currently there is only the work of Ragone et al. published, which examines the vaccination response for a possible affiliation to specific HLA alleles [44]. With regard to our data, an association between certain HLA alleles and a reduced vaccination response could not be shown. However, it has to be considered that only a small number of patients were examined. The work of Ragone et al. studied 56 health care workers, and demonstrated that short- and medium-term antibody response to the COVID-19 vaccine does not correlate with the HLA class II allele [44]. This would mean that a reduced vaccination response, exclusively linked to individual characteristics, is not related to the HLA background. Based on this, a follow-up study with a detailed HLA typing of the study participants would certainly be useful. In particular, the question of whether there is an association between HLA alleles and SARS-CoV-2-infection breakthroughs after successful vaccination would be of great interest.

The limitations of our work are the missing measurements of neutralizing autoantibodies investigating the humoral vaccination response. However, there is increasing evidence that anti-spike antibody titers correlate with neutralizing capacity [33]. In addition, with regard to the antigen-specific cellular vaccination response, we only determined SARS-CoV2 antigen-specific cells using an IFN-γ/IL-2 ELISpot assay. However, a general statement about how effectively these cells actually work in controlling a SARS-CoV-2 infection cannot be made with these results. Furthermore, the small number of patients in whom we investigated a possible link between the HLA genotype and the vaccination response should be mentioned.

## 4. Materials and Methods

### 4.1. Study Design

The study was initiated to investigate the humoral and antigen-specific cellular vaccination response in HDP and KTR compared to a healthy control group. The ethics committee of the University Medical Centre Mainz, Germany and the medical association of the state of Rhineland Palatinate approved this study (Approval No 2021-15786). The study was conducted in accordance with the Declaration of Helsinki of the World Medical Association. 

HDP patients were recruited from the outpatient hemodialysis unit of our tertiary care center as well as from two private practices. KTRs were recruited from outpatient clinic. Written informed consent was obtained from all participants. All patients were vaccinated twice with BNT162b2 (Pfizer-BioNTech, Mainz, Germany) with an interval of 4 weeks. Blood samples were taken 1 month after the first vaccination, 1 month after the second vaccination, and 6 months after the first vaccination, as indicated in the study’s flow chart (Figure 3). The number of patients in relation to each study visit can be observed in Appendix A.

Blood samples for antibody determination were aliquoted and stored at −30 °C and then analyzed collectively. Whole blood samples for the analysis of T-cell responses were processed immediately.

### 4.2. Isolation of Peripheral Blood Mononuclear Cells (PBMCs)

PBMCs were isolated from 20 mL of heparinized blood by density gradient centrifugation using Pancoll human solution (PAN-Biotech GmbH, Aidenbach, Germany). First, the blood was diluted 1.5× with phosphate-buffered saline (PBS). This mixture was gently pipetted onto the Pancoll solution. The gradient was immediately centrifuged for 30 min with 1000× *g* without breaks. The plasma was discarded, and the mononuclear cells layer was transferred into a new tube, washed twice with PBS and finally once with AIM-V medium 10 min at 700× *g*. Then, cells were counted and adjusted with AIM-V medium to a concentration of 2 × 10^6^ cells per ml.

### 4.3. ELISpot Assay

To assess the SARS-CoV-2-specific cellular response, we performed an IFN-γ/IL-2 ELISpot assay using the CoV-*i*Spot IFN-γ + IL-2 assay in strip format (AID Diagnostics, Straßberg, Gemany). Briefly, a 96-well plate was precoated with monoclonal antibodies against either IFN-γ or IL-2. A total of 100 µL of the medium, Pokeweed mitogen (PWM), or SARS-CoV-2 antigen solution were added in duplicates. The PBMC cell suspension was mixed with anti-human CD28 (1:1000), and 100 µL of this mixture were added to each well. The plate was incubated for 20 to 24 h at 37 °C and 5% CO_2_. Afterwards, the supernatent containing the cells was removed and several washing steps were applied. Then, a biotin-conjugated and a FITC-labeled antibody against IFN-γ and IL-2 were added and multiplexed. After 2 h of incubation, the wells were washed again, and streptavidin RED-Cy3 and anti-FITC green were added for 1 h. After more washing steps, a proprietary enhancer solution was added for 15 min. The plate was dried overnight and protected from the light. Spot enumeration was performed with the AID *i*Spot Reader System (Autoimmune Diagnostica GmbH, Straßberg, Germany). A stimulation index (SI) was computed by dividing the mean number of spots in an antigen-containing well through the mean number of spots in the negative control. Stimulated spot numbers >7-fold higher (if negative control 0–1 spot) or >3-fold higher (if negative control 2–20 spots) than the negative control were considered positive. Either a positive IFN-γ or a positive IL-2 result was considered indicative of a T-cell-specific immune response to vaccination.

### 4.4. Detection of Antibodies

In order to exclude patients with prior SARS-CoV-2 infection, a qualitative chemiluminescence microparticle immunoassay was used for the qualitative detection of IgG antibodies against the SARS-CoV-2 nucleocapsid protein (Architect SARS-CoV-2-IgG, Abbott GmbH, Wiesbaden, Germany). 

A quantitative analysis of IgG antibodies directed against the receptor-binding domain (RBD) of the SARS-CoV-2 spike protein was performed by chemiluminescence microparticle immunoassay (CMIA) (Architect SARS-CoV-2-IgG II Quant, Abbott GmbH, Wiesbaden, Germany). Vaccine response was considered to be present at a titer of >50 AU/mL (the threshold specified by the manufacturer).

### 4.5. Statistics

Data are presented as means ± standard error of the means (SEM). The nonparametric Mann–Whitney U test was employed to assert statistical significance and multiple comparisons were analyzed using the Kruskal–Wallis test. For correlation analysis, we used the Spearman correlation calculation. A *p* value below 0.05 was considered to indicate statistical significance. All analyses were carried out with GraphPad Prism version 7.0 (GraphPad, San Diego, CA, USA).

## 5. Conclusions

In conclusion, mRNA-based vaccines are safe and effective in HDP with a strong humoral as well as antigen-specific cellular vaccination response after the second vaccination with BNT162b2, but with a significant anti-spike antibody titer reduction after 6 months. However, based on the recommended booster vaccination, this population should continue to be well protected against severe SARS-CoV-2 infections, in particular. In contrast, a significantly weaker humoral as well as antigen-specific cellular vaccination response is observed in KTR. Due to this, it is mandatory to elaborate concepts for an improvement of vaccination response in KTR. Here, in the future, the switch between different vaccination mechanisms (mRNA-based vaccines, protein subunit-based vaccines and whole-inactivated vaccines) may become increasingly important. Until further options for KTR may be available, repeated vaccination is the most advisable with mRNA-1273 (Moderna) and is essential in providing the highest possible protection against severe SARS-CoV-2 courses in this population.

## Figures and Tables

**Figure 1 pathogens-11-00067-f001:**
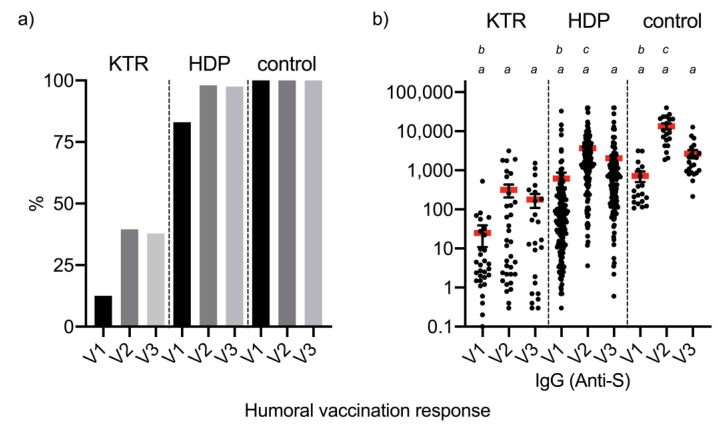
Humoral vaccine response measured by SARS-CoV-2-specific antibodies. (**a**) Shows the vaccination response in percentage measured by anti-spike (s) IgG antibodies at 1 month after the 1st vaccination (V1), 1 month (V2) after the 2nd vaccination, and 6 months (V3) after 1st vaccination. The x-axis represents the time points of measurement of vaccination response; the y-axis represents the percentage. (**b**) Shows the vaccination response measured by anti-spike (s) IgG antibodies in AU/mL at V1, V2, and V3. The vaccination response is divided into 3 study groups: patients after kidney transplantation (KTR), dialysis patients (HDP), and healthy controls (control). The x-axis represents in (**b**) the time points of measurement of vaccination response; the y-axis represents the anti-spike IgG antibody titer in AU/mL. The italicized letters present the statistical analysis: a = present the statistical significance between the corresponding time points in the different study groups; b = demonstrates statistical significance of V1 to V2 within the study group; c = demonstrates statistical significance of V2 to V3 within the study group.

**Figure 2 pathogens-11-00067-f002:**
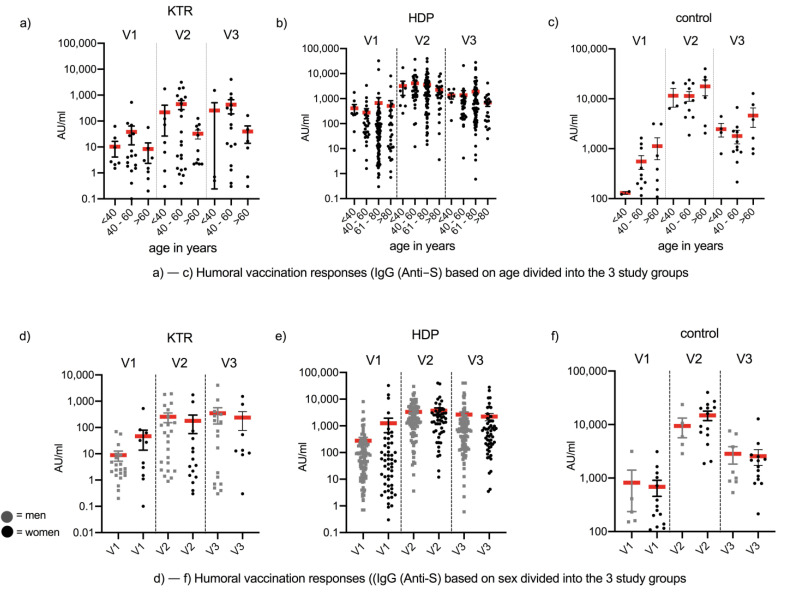
Humoral vaccine response measured by SARS-CoV-2-specific antibodies referred to age and sex. (**a**–**c**) Based on age, the humoral vaccination response in each of the 3 study groups is shown at 1 month after the 1st vaccination (V1), 1 month (V2) after 2nd vaccination, and 6 months (V3) after the 1st vaccination. The x-axis represents age groups (**a**,**c**) <40y, 40–60, >60; (**b**) <40y, 40–60, 61–80, >80) of the testing; the y-axis represents the SARS-CoV-2 autoantibody (IgG anti-S) titer in AU/mL. (**d**–**f**) (Grey = male; black = female)**.** Based on sex, the humoral vaccination response in each of the 3 study groups is shown at 1 month after the 1st vaccination (V1) and 1 month (V2) after the 2nd vaccination. The x-axis represents the time points of the testing; the y-axis represents the SARS-CoV-2 autoantibody (IgG anti-S) titer in AU/mL.

**Figure 3 pathogens-11-00067-f003:**
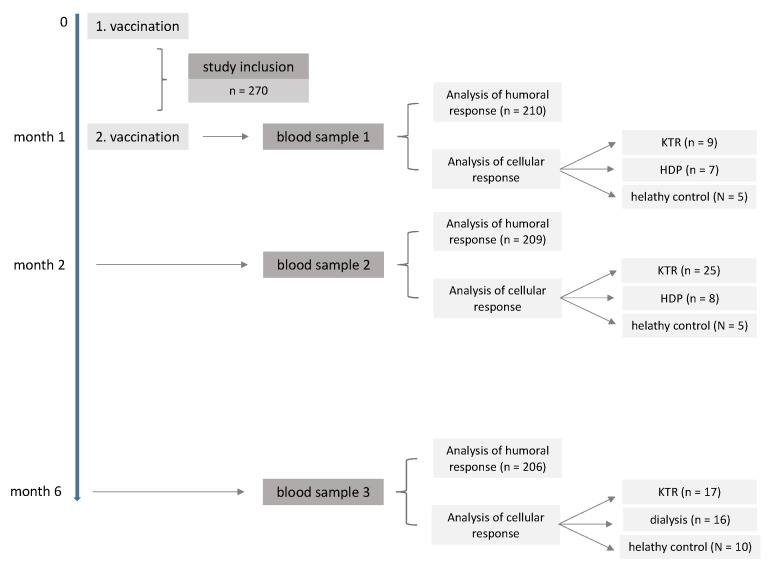
Study’s flow chart. Shown is the study design with the time of vaccination and measurement of the vaccination response divided into the cellular and humoral vaccination response.

**Table 1 pathogens-11-00067-t001:** Patient characteristics.

	HDP (*n* = 192)	KTR (*n* = 50)	Control (*n* = 28)
**Age (in years)** with SD	66.9 (±14.7)	51.2 (±13.8)	54.1 (±12.2)
**Sex** f/m (f/m in %)	66/126 (34.5/66)	18/32 (36/64)	19/9 (68/32)
**Comorbidities (in %)**			
Overweight (BMI 26–31)	50 (29.8)	15 (31.9)	9 (36)
Obesity (BMI > 31)	36 (21.4)	6 (12,8)	1 (4)
Diabetes	51 (30.4)	14 (29.8)	1 (4)
Hypertension	119 (70.8)	33 (70.2)	5 (20)
Heart disease	79 (47)	14 (29.8)	1 (4)
Chronic lung disease	25 (14.9)	4 (8.2)	0
Gastrointestinal/liver disease	21 (12.5)	9 (19.2)	1 (4)
Tumor disease	33 (19.6)	3 (6.4)	1 (4)
**Smoking**			
PY (pack years)	28.3 (±14.7)	17 (±10.3)	22 (±4)
Active smoker	17	1	3
Former smoker	33	13	1

**Table 2 pathogens-11-00067-t002:** Demographics of the kidney transplantation cohort.

Transplantation Data	
**Years since transplantation**	(*n* = 50)
<3	20
3–6	7
7–11	9
>12	11
**Immunosuppression**	
Glucocorticoid	49 (3.74 mg (±1.5)
Calcineurin inhibitor	45
MMF	40
mTOR inhibitor	8
other	2
**Immunosuppression regime**	
Dual immunosuppression	5
Triple immunosuppression	45

**Table 3 pathogens-11-00067-t003:** Measurements of SARS-CoV-2 antigen-specific cellular response. Shown is the cellular vaccination response measured by IL-2 and IFN-γ-producing cells 1 month after the 1st vaccination, 1 month after the 2nd vaccination, and 6 months after the 1st vaccination.

	**Reactivity**	**V1 (*n* = 7)**	**V2 (*n* = 8)**	**V3 (*n* = 16)**
**Dialysis patients**	Only IFN-γ	0%	0%	6.25%
	Only Il-2	0%	25%	18.75%
	IFN-γ and IL-2	29%	62%	43.74%
	No detection	71%	13%	31.25%
		**V1 (*n* = 9)**	**V2 (*n* = 25)**	**V3 (*n* = 17)**
**Kidney transplant recipients**	Only IFN-γ	11%	0%	0%
	Only IL-2	22%	12%	41%
	IFN-γ and IL-2	0%	48%	23.5%
	No detection	67%	40%	35.5%
		**V1 (*n* = 5)**	**V2 (*n* = 5)**	**V3 (*n* = 10)**
**Healthy control**	Only IFN-γ	0%	0%	0%
	Only IL-2	60%	0%	30%
	IFN-γ and IL-2	40%	100%	60%
	No detection	0%	0%	10%

## Data Availability

All data are provided by the corresponding author.

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
