# Peer review of "Six-Month Follow-Up after Vaccination with BNT162b2: SARS-CoV-2 Antigen-Specific Cellular and Humoral Immune Responses in Hemodialysis Patients and Kidney Transplant Recipients"

_pathogens, 2022, doi:10.3390/pathogens11010067_

Round 1

Reviewer 1 Report

Summary:

In this study, the authors monitored the humoral and cellular immune responses against SARS-CoV-2 antigen at 1, 2, and 6 months after the first dose of BNT162b2 mRNA vaccine in hemodialysis patients (HDP) and kidney transplant recipients (KTR). Their results showed that, compared to healthy control, HDPs and KTRs have weaker humoral and cellular immune responses. There are some minor concerns that need to be addressed.

  1. Lines 28-29, to help readers understand the data, I would suggest changing the order of “HDP, HC, and KTR” to “HDP, KTR, and HC” to keep consistent with other places in the Abstract.
  2. There are some typos in the manuscript: line 157, “Figure 2b” should be “Figure 1b”; lines 220-221, “Figure 3b” should be “Figure 2b”; line 226, “3c” should be “2c”; lines 247-248, “supplement Table 2 and 3” should be “supplement Table 1 and 2”; line 291, “INF” should be “IFN”.
  3. To better present the data, I would suggest adding more points in the y-axis to Figure 1a, at least include 25% and 75%. In addition, the authors may consider using superscript letters to present the statistical analysis results in Figure 1b, instead of drawing lines between each pair of data.
  4. In section 2.3., the authors should indicate which figure/table did they refer to at an early time, not at the end of the paragraph.
  5. Lines 202-208, it is not clear to me that how did the authors calculate 72% (line 204) and 70.5% (line 206). Also, did the authors perform the same analysis for HDP patients?

Author Response

Lines 28-29, to help readers understand the data, I would suggest changing the order of “HDP, HC, and KTR” to “HDP, KTR, and HC” to keep consistent with other places in the Abstract.

Thank you for this comment. We changed the order of the study groups as you suggested.

There are some typos in the manuscript: line 157, “Figure 2b” should be “Figure 1b”; lines 220-221, “Figure 3b” should be “Figure 2b”; line 226, “3c” should be “2c”; lines 247-248, “supplement Table 2 and 3” should be “supplement Table 1 and 2”; line 291, “INF” should be “IFN”.

Thank you for your correction. We have resolved the typos.

To better present the data, I would suggest adding more points in the y-axis to Figure 1a, at least include 25% and 75%. In addition, the authors may consider using superscript letters to present the statistical analysis results in Figure 1b, instead of drawing lines between each pair of data.

Thank you for your suggestions. We added more points to the y-axis in Figure 1a and we implemented italicized letters to present the statistical analysis in Figure 1b.

In section 2.3., the authors should indicate which figure/table did they refer to at an early time, not at the end of the paragraph.

Thank you for your comment. We referred table 3 to an early time.

Lines 202-208, it is not clear to me that how did the authors calculate 72% (line 204) and 70.5% (line 206). Also, did the authors perform the same analysis for HDP patients?

Thank you for your comment. 60% of the KTR patients (see table 3) presented a SARS-CoV-2 antigen specific cellular response in V2. In addition, an exclusively humoral vaccination response was demonstrated in 12 % of patients in V2, resulting in a total of 72% of patients presenting with an overall vaccination response. Similarly, the 70,5 can be calculated. In dialysis patients, we have not listed this again, because almost 100% showed a humoral vaccination response.

Reviewer 2 Report

In this study, Boedecker-Lips et al. report the humoral and cellular immune responses to Pfizer/BNT CoVID19 mRNA vaccine in dialysis and kidney transplanted patients. Although a few similar studies have been conducted in immunosuppressed/immunocompromised cohorts, this study is a valuable addition. Overall, the study was designed properly and carried out and the manuscript is well written. This reviewer has a few minor comments for the authors’ consideration.

  1. When testing the responsiveness of patients to vaccination, the individual patient was assayed at different times (V1, V2, and V3). These datasets of the same patient should be considered as matched pairs. Therefore, the Mann-Whitney U test may underestimate the significance of the difference and may not be the best test. The McNemar’s Test or a logistic regression model for ordinal matched-pairs responses may be considered.
  2. Line 43, “Antibody data suggests that the immune response to SARS-CoV-2 vaccination”, references should be provided for this statement.
  3. Typos: line 49: patients; line 59: trial
  4. Line 250: “anti-spike antibodies against the nucleocapsid protein”, please clarify this sentence.

Author Response

When testing the responsiveness of patients to vaccination, the individual patient was assayed at different times (V1, V2, and V3). These datasets of the same patient should be considered as matched pairs. Therefore, the Mann-Whitney U test may underestimate the significance of the difference and may not be the best test. The McNemar’s Test or a logistic regression model for ordinal matched-pairs responses may be considered.

The reviewer’s suggestion to perform a qualitative analysis is greatly appreciated and we have thoroughly considered performing a logistic regression. However, a positive response is not clearly defined (as discussed in our manuscript). While all of the healthy control subjects had a substantial increase in antibody titers, consistent with 100% response rate, many dialysis and transplant patients had only modest increases in antibody levels. This is currently described and discussed in the paper. Reducing the data to yes/no responses would obscure this detail. Therefore, in our humble opinion, we would rather refrain from performing the logistic analysis, and we hope that the reviewer can subscribe to this reasoning.

Line 43, “Antibody data suggests that the immune response to SARS-CoV-2 vaccination”, references should be provided for this statement.

Thank you for this comment. We implemented this recommendation and added references.

Typos: line 49: patients; line 59: trial

Thank you for your correction. We have resolved the typos.

Line 250: “anti-spike antibodies against the nucleocapsid protein”, please clarify this sentence.

Thank you for this comment. We have rephrased the sentence.